# Interdependent IL-7 and IFN-γ signalling in T-cell controls tumour eradication by combined α-CTLA-4 + α-PD-1 therapy

Lewis Zhichang Shi[1,*], Tihui Fu[1,*], Baoxiang Guan[1], Jianfeng Chen[1], Jorge M. Blando[2], James P. Allison[2,3], Liangwen Xiong[1], Sumit K. Subudhi[1], Jianjun Gao[1] & Padmanee Sharma[1,2,3]

Combination therapy with α-CTLA-4 and α-PD-1 has shown significant clinical responses in different types of cancer. However, the underlying mechanisms remain elusive. Here, combining detailed analysis of human tumour samples with preclinical tumour models, we report that concomitant blockade of CTLA-4 and PD-1 improves anti-tumour immune responses and synergistically eradicates tumour. Mechanistically, combination therapy relies on the interdependence between IL-7 and IFN-γ signalling in T cells, as lack of either pathway abrogates the immune-boosting and therapeutic effects of combination therapy. Combination treatment increases IL-7Rα expression on tumour-infiltrating T cells in an IFN-γ/IFN-γR signalling-dependent manner, which may serve as a potential biomarker for clinical trials with immune checkpoint blockade. Our data suggest that combining immune checkpoint blockade with IL-7 signalling could be an effective modality to improve immunotherapeutic efficacy. Taken together, we conclude that combination therapy potently reverses immunosuppression and eradicates tumours via an intricate interplay between IFN-γ/IFN-γR and IL-7/IL-7R pathways.

[1] Department of Genitourinary Medical Oncology, MD Anderson Cancer Center, Houston, Texas 77030, USA. [2] The Immunotherapy Platform, MD Anderson Cancer Center, Houston, Texas 77030, USA. [3] Department of Immunology, MD Anderson Cancer Center, Houston, Texas 77030, USA. * These authors contributed equally to this work. Correspondence and requests for materials should be addressed to P.S. (email: PadSharma@mdanderson.org).

T-cell activation in response to T cell receptor (TCR) ligation and CD28 co-stimulation is counter-balanced by induction of a group of inhibitory receptors and ligands, known as 'immune checkpoints', to limit collateral tissue damage during anti-microbial immune responses. CTLA-4 and PD-1 are the first immune checkpoints to be characterized and clinically targeted[1–3]. However, these checkpoints may also diminish anti-tumour immune responses. Thus, blocking these checkpoints represents a legitimate approach to boost tumour immunity. We previously found that α-CTLA-4 blockade inhibits tumour growth and promotes tumour-free survival of tumour-bearing mice[4], contributing to the recent FDA approval of ipilimumab, a human α-CTLA-4 monoclonal antibody that improves overall survival in patients with metastatic melanoma[5,6]. These findings, together with recent reports that blocking the PD-1/PD-L1 pathway improves overall survival and objective responses in patients with metastatic melanoma[7,8], helped to establish a new field of 'immune checkpoint blockade'. Currently, immune checkpoint therapy is considered as a standard treatment for patients with some types of cancer including advanced melanoma, non-small cell lung cancer and metastatic kidney cancer. Nevertheless, only a fraction of these patients respond to immune checkpoint therapy. Ongoing efforts are focusing on novel strategies to improve the efficacy.

Combination therapy with α-CTLA-4 and α-PD-1 has shown strong anti-tumour immune responses in preclinical murine melanoma[9], murine CT26 colon carcinoma and ID8-VEGF ovarian carcinoma[10] and metastatic osteosarcoma[11]. Improved therapeutic effects of combination therapy have also been demonstrated in patients with advanced melanoma[12,13]. Promising preliminary results of combination therapy in patients with renal cell carcinoma (RCC)[14] or with non-small cell lung cancer[15] were recently reported. Moreover, combination therapy was initiated for patients with other advanced solid tumours including select gastrointestinal cancers, head and neck squamous cell carcinoma, and hepatocellular carcinoma[16]. These reports highlight combination therapy as an effective strategy to improve therapeutic efficacy. Despite these promising results, the underlying mechanisms for combination therapy are largely unknown.

Building on the first preoperative clinical trial of α-CTLA-4 treatment in patients with urinary bladder cancer[17], we attempted to elucidate the underlying mechanisms of combination therapy-mediated tumour rejection by performing detailed analysis of human bladder tumour samples together with preclinical studies using the murine MB49 bladder tumour model, which shares impressive similarities with human bladder cancer including cell surface markers, sensitivity to apoptosis and immunological profiles[18,19]. We found that combination therapy-improved tumour rejection by promoting T-cell infiltration into tumours, proliferation and polyfunctionality of tumour-infiltrating lymphocytes (TILs), and expansion of endogenous memory T cells, which are mediated by the interdependent loop between IL-7 and IFN-γ signalling in T cells. We provided direct evidence that additional blockade of α-PD-1 overcame tumour 'escape' from α-CTLA-4 monotherapy and resulted in complete tumour rejection with long-lasting protective immunity to re-challenge, which is predominantly T-cell-dependent and natural killer (NK)/natural killer T (NKT) cell-independent.

## Results

**α-CTLA upregulates PD-1/PD-L1 inhibitory pathway.** Our first surgical clinical trial of α-CTLA-4 in patients with bladder cancer detected clinical signals in only 3 out of 12 patients[17], suggesting existence of other important suppressive mechanisms. The PD-1/PD-L1 pathway is a primary ligand–receptor coinhibitory interaction in tumours[20]. To examine if the PD-1/PD-L1 pathway can be attributed to the low efficiency of α-CTLA-4 monotherapy, we analyzed PD-1 and CTLA-4 expression on TILs isolated from human and murine bladder tumours. While TILs from human bladder tumour predominantly co-expressed CTLA-4 and PD-1 (Fig. 1a), 25% of TILs from murine MB49 tumours co-expressed CTLA-4 and PD-1 with additional 36 and 4% expressing either PD-1 or CTLA-4 alone (Fig. 1b). We also detected high expression of PD-L1 on human (Fig. 1c) and murine bladder tumour cells (Fig. 1d). These results indicate PD-1/PD-L1 could represent an important immunosuppressive pathway in bladder tumours.

We then assessed how α-CTLA-4 impacts the PD-1/PD-L1 pathway in human and murine bladder tumours. Consistent with our previous report using an aggressive murine melanoma model[9], we observed increased PD-1 density on immune cells in human bladder tumour samples after anti-CTLA-4 therapy (Fig. 1e). Similarly, we detected increased PD-1 expression on CD4$^+$Foxp3$^-$ TILs isolated from α-CTLA-4-treated MB49 tumour-bearing mice (Fig. 1f). An increasing trend of PD-1 expression on CD8$^+$ TILs cells was also noticed (Supplementary Fig. 1A). Moreover, PD-L1 expression was higher on human bladder tumour cells (Fig. 1g) and immune cells (Fig. 1h) treated with α-CTLA-4. Collectively, our analyses of bladder tumours reveal co-expression of PD-1 and CTLA-4 on TILs, high expression of PD-L1 on tumour cells and augmentation of PD-1/PD-L1 inhibitory pathway by α-CTLA-4, suggesting simultaneous blockade with α-CTLA-4 and α-PD-1 is required for rejection of bladder tumours.

**Eradication of bladder tumours requires combination therapy.** To test this idea, we treated MB49-bearing C57BL/6 wild-type (WT) male mice with α-CTLA-4, α-PD-1 or combination of both. Male mice were used throughout this study to avoid interfering gender mismatch effects on immunological responses, as MB49 was derived from a male C57BL/6 mouse[21]. To evaluate therapeutic effects, antibodies were given on day 6 post-tumour inoculation when palpable tumours appeared. While α-PD-1 only delayed tumour growth, α-CTLA-4, on the other hand, protected ∼40% of mice from tumour challenge (Fig. 2), similar to what was observed in patients with bladder cancer treated with α-CTLA-4 (ref. 17). In contrast, combination therapy eradicated tumours in 100% of the mice (Fig. 2a,b), confirming that concomitant blockade with α-CTLA-4 and α-PD-1 is required for effective rejection of bladder tumours. Interestingly, although α-CTLA-4 monotherapy initially suppressed tumour growth in all the mice, tumours in ∼60% of mice escaped this early suppression and resumed growth, eventually leading to death of these mice (Fig. 2a). When α-PD-1 was added to α-CTLA-4 blockade as in the combination group, 'escape' of tumour from α-CTLA-4 monotherapy was blocked and all tumours were rejected, providing direct evidence that MB49 tumours utilize the PD-1 pathway to evade α-CTLA-4 monotherapy. Clearly, these results point to synergistic effects of α-CTLA-4 and α-PD-1 in eliminating bladder tumours.

We further confirmed the synergistic effects of combination therapy using a different murine tumour model, the RENCA model for RCC, where ∼70% of RENCA-bearing mice survived in combination therapy group and only 30% of or 0% of mice survived in α-CTLA-4 or α-PD-1 monotherapy group, respectively (Supplementary Fig. 1B,C).

**Combination therapy requires T cells for tumour rejection.** Next, we asked which immune cells contribute to therapeutic effects of combination therapy. Since both α-CTLA-4 and α-PD-1

treatment increase tumour T-cell infiltration in mouse and human[22–24], we examined the abundance of CD4$^+$ and CD8$^+$ T cells in MB49 tumours treated with α-CTLA-4, α-PD-1 or combination therapy. While α-CTLA-4 or α-PD-1 monotherapy induced CD4$^+$ and CD8$^+$ T-cell infiltration into tumours, combination therapy resulted in markedly greater infiltration of both (Fig. 3a). Consistent with the strong depleting effects of α-CTLA-4 on intratumoral $T_{reg}$[25], we observed a dramatic reduction of $T_{reg}$ in MB49 tumours treated with either α-CTLA-4 or combination therapy, with the latter being slightly stronger (Fig. 3b). Consequently, combination therapy generated the greatest CD8$^+$/$T_{reg}$ and CD4$^+$ non-$T_{reg}$/$T_{reg}$ ratios (Fig. 3c,d). To address if combination therapy-mediated increases of CD4$^+$ and CD8$^+$ T cells are responsible for improved efficacy of combination therapy in bladder tumour, MB49 tumour-bearing mice were pretreated with specific CD4$^+$ and/or CD8$^+$ T-cell depleting antibodies before combination therapy. Depletion of either CD4$^+$ (GK1.5) or CD8$^+$ (2.43) T cells abolished mouse survival induced by combination therapy, although tumour growth was still delayed (Fig. 3e). As a comparison, deletion of both CD4$^+$ and CD8$^+$ T cells (GK1.5 + 2.43) ablated both improved mouse survival and suppressed tumour growth (Fig. 3e) by combination therapy,

pointing to critical roles of CD4$^+$ and CD8$^+$ T cells in this process.

In addition to CD4$^+$ and CD8$^+$ T cells, NK cell activity has long been associated with the destruction of tumour cells both *in vitro* and *in vivo*[26]. More recently, a specialized subset of T cells was found to coexpress NK cell markers (for example, NK1.1), namely NKT cells[27], which also showed natural tumour surveillance capabilities[28]. Therefore, we explored the potential involvement of NK/NKT cells by analyzing abundance of NK and NKT cells in MB49-bearing mice treated with mono- or combination therapy. Our results indicate neither α-CTLA-4 nor α-PD-1 affects distribution of NK and NKT cells in MB49 bladder tumours (Supplementary Fig. 2A). To further elucidate the role of NK and NKT cells in combination therapy, we deleted NK1.1$^+$ NK and NKT cells using α-NK1.1 mAb PK136 before combination treatment, which did not affect beneficial effects of combination therapy (Supplementary Fig. 2B), indicating NK and/or NKT cells are not required. Taken together, our data show that CD4$^+$ and CD8$^+$ T cells but not NK and NKT cells are the prominent mediators for combination therapy of bladder cancer.

**Combination therapy increases polyfunctionality of TILs.** Next, we attempted to understand the underlying molecular

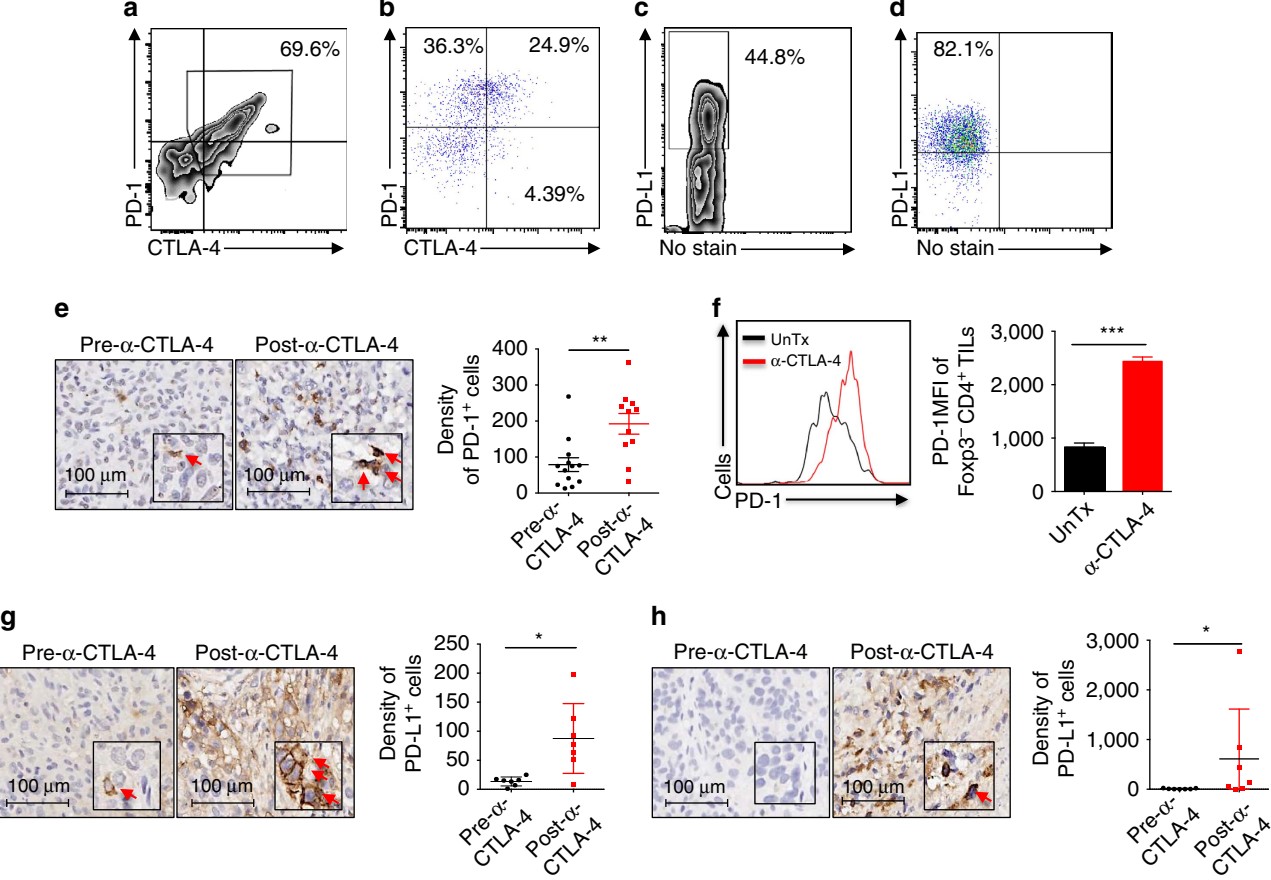

**Figure 1 | α-CTLA-4 upregulates the PD-1/PD-L1 inhibitory pathway in human and murine bladder tumours.** (a,b) Surface expression of CTLA-4 and PD-1 in Foxp3$^-$CD4$^+$ TILs isolated from patients with bladder cancer (**a**) and from murine MB49 bladder tumours (**b**). (**c,d**) PD-L1 expression on human (**c**) and mouse (**d**) bladder tumour cells analyzed by flow cytometry. (**e**) PD-1 expression on immune cells from pre- and post-α-CTLA-4-treated human bladder tumour by IHC. Representative IHC photos (arrows indicate positively-stained cells) and pooled results from all the patients are shown. (**f**) PD-1 expression on Foxp3$^-$CD4$^+$ TILs isolated from MB49 tumour-bearing mice treated with (α-CTLA-4) or without α-CTLA-4 (UnTx) were analyzed by flow cytometry. Histograms of PD-1 (left) and pooled results (right) from five mice in one representative experiment are shown. (**g,h**) PD-L1 expression on tumour cells (**g**) and immune cells (H) of pre- and post-α-CTLA-4-treated human bladder samples by IHC. Representative IHC photos (arrows indicate positively-stained cells) and pooled results from all the matched cases are shown. Data in scatter plots and the bar graph are means ± s.e.m. Representative results (**b**, **d**, and **f**) from three independent experiments are shown. *$P < 0.05$; **$P < 0.01$; ***$P < 0.001$ by two-tailed unpaired Student's $t$-test.

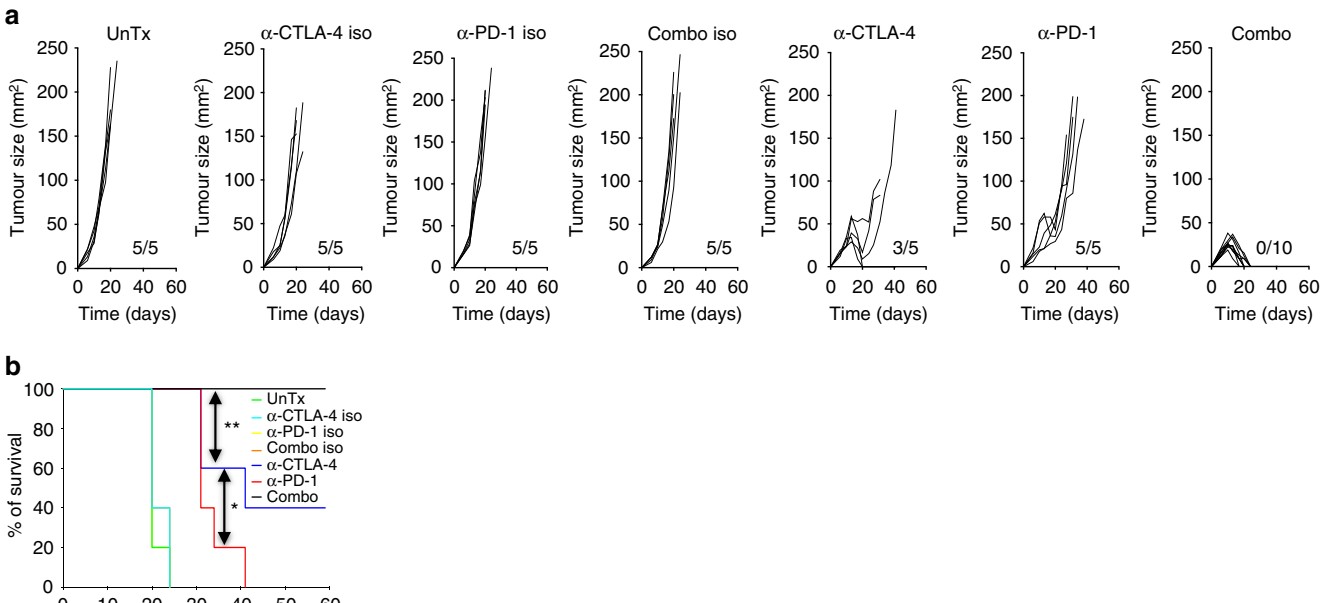

**Figure 2 | Combined PD-1 and CTLA-4 blockade eradicates MB49 bladder tumours in mice.** Mice bearing 6 days palpable MB49 tumour were left untreated (UnTx) or treated with α-CTLA-4, α-PD-1 or combination of both (Combo) or their corresponding isotype controls (α-CTLA-4 iso, α-PD-1 iso or Combo iso). (**a**) Individual tumour growth. Tumour size is presented as length × width (mm$^2$) and ratios of tumour-growing mice in each groups are shown as insets on each panel. (**b**) Survival curves from one representative experiment are shown. Data are representative of three independent experiments. *$P<0.05$; **$P<0.01$; ***$P<0.001$ by log-rank (Mantel–Cox) test.

mechanisms for the combination therapy. Since combination therapy significantly improved CD4$^+$ non-$T_{reg}$/$T_{reg}$ and CD8$^+$/$T_{reg}$ ratios as compared to α-CTLA-4 or α-PD-1 monotherapy in tumours (Fig. 3c,d), we reasoned that combination therapy could more potently stimulate effector functions of TILs. First, we co-cultured purified CD4$^+$ or CD8$^+$ TILs together with irradiated MB49 tumour cells and splenic dendritic cells (DCs) to evaluate production of tumour antigen-specific effector molecules by TILs. As expected, we observed increased production of IFN-γ by α-CTLA-4 and addition of α-PD-1 further boosted IFN-γ production to a significantly higher level in CD4$^+$ TILs (Fig. 4a). Additional detailed analysis of IFN-γ$^+$ cells revealed that the greatest increase induced by combination therapy was IFN-γ$^+$IL-2$^+$ dual producers (~20–40-fold, Fig. 4b), while α-PD-1 only had minimal effect in inducing these dual producers. Similarly, a greater increase of IFN-γ$^+$IL-2$^+$ dual producers was noticed in CD8$^+$ TILs treated with combination therapy (Supplementary Fig. 3A), correlated with its greater therapeutic efficacy. Anti-tumour activity of both IL-2 and IFN-γ has long been recognized[29,30]. Considering IL-2 is required for long-term survival of tumour antigen-specific T cells[31], we reason that these dual producers possess inherent advantages to survive the hostile tumour microenvironment (TME). Second, we briefly stimulated TILs with phorbol-12-myristate 13- acetate (PMA) and ionomycin to measure production of IFN-γ and IL-2. Similar to the above results, we detected significant increase of IFN-γ$^+$IL-2$^+$ dual producers in CD4$^+$ and CD8$^+$ TILs treated with combination therapy (Supplementary Fig. 3B,C). It was suggested that TILs expressing both IFN- γ and TNF-α are the most potent effectors[32]. In support of this, we found IFN-γ and TNF-α double-positive CD4$^+$ (Fig. 4c,d) and CD8$^+$ TILs (Fig. 4e,f) were most highly represented following combination therapy.

In addition to increased production of effector cytokines, another important hallmark for tumour T-cell reinvigoration is the restoration of their proliferative ability, which can be evaluated with Ki-67 expression. To simultaneously assess both aspects of immunosuppression reversal by combination therapy, Ki-67 staining was conducted together with IFN-γ/production in re-stimulated TILs. Significant increases of Ki-67$^+$IFN-γ$^+$ cells were seen in both CD4$^+$ TILs and in CD8$^+$ TILs treated with combination therapy (Supplementary Fig. 3D,E). Collectively, combination therapy enhances proliferative polyfunctional TILs.

**Combination therapy expands memory cells via IL-7R signalling.** A cardinal feature of immune checkpoint therapy is the induced durability of tumour rejection responses[2]. Therefore, we assessed how combination therapy affects formation of endogenous memory response. Since memory cells upon formed primarily reside in the spleen and secondary lymphoid organs, we characterized splenocytes isolated from MB49 tumour-bearing mice with regressing tumours, a stage when memory T cells are generated. We directed our attention to the central memory population of CD4$^+$ and CD8$^+$ T cells (CD44$^+$CD122$^{hi}$, $T_{CM}$), as these cells are long-lived and likely mediate protective immunity to re-challenge[33]. As shown in Fig. 5a,b, while α-CTLA-4 or α-PD-1 alone did not significantly expand endogenous CD4$^+$ and CD8$^+$ $T_{CM}$ cells, combination therapy increased both proportions and absolute cell numbers of $T_{CM}$ cells, as compared to the untreated mice. This could explain why the majority of mice treated with α-CTLA-4 or α-PD-1 monotherapy delayed tumour growth but could not completely eradicate already-formed tumours (Fig. 2). To evaluate if combination therapy-induced expansion of endogenous memory T-cell pool boosts tumour antigen-specific memory response, we re-challenged combination therapy-treated mice that had cleared primary MB49 tumours with 10-fold more MB49 tumour cells or unrelated B16/BL6 melanoma cells. Clearly, combination therapy-treated mice were resistant to re-challenge with the same MB49 tumour cells, but remained vulnerable to unrelated B16/BL6 tumour inoculation (Fig. 5c), indicating formation of tumour antigen-specific protective immunity.

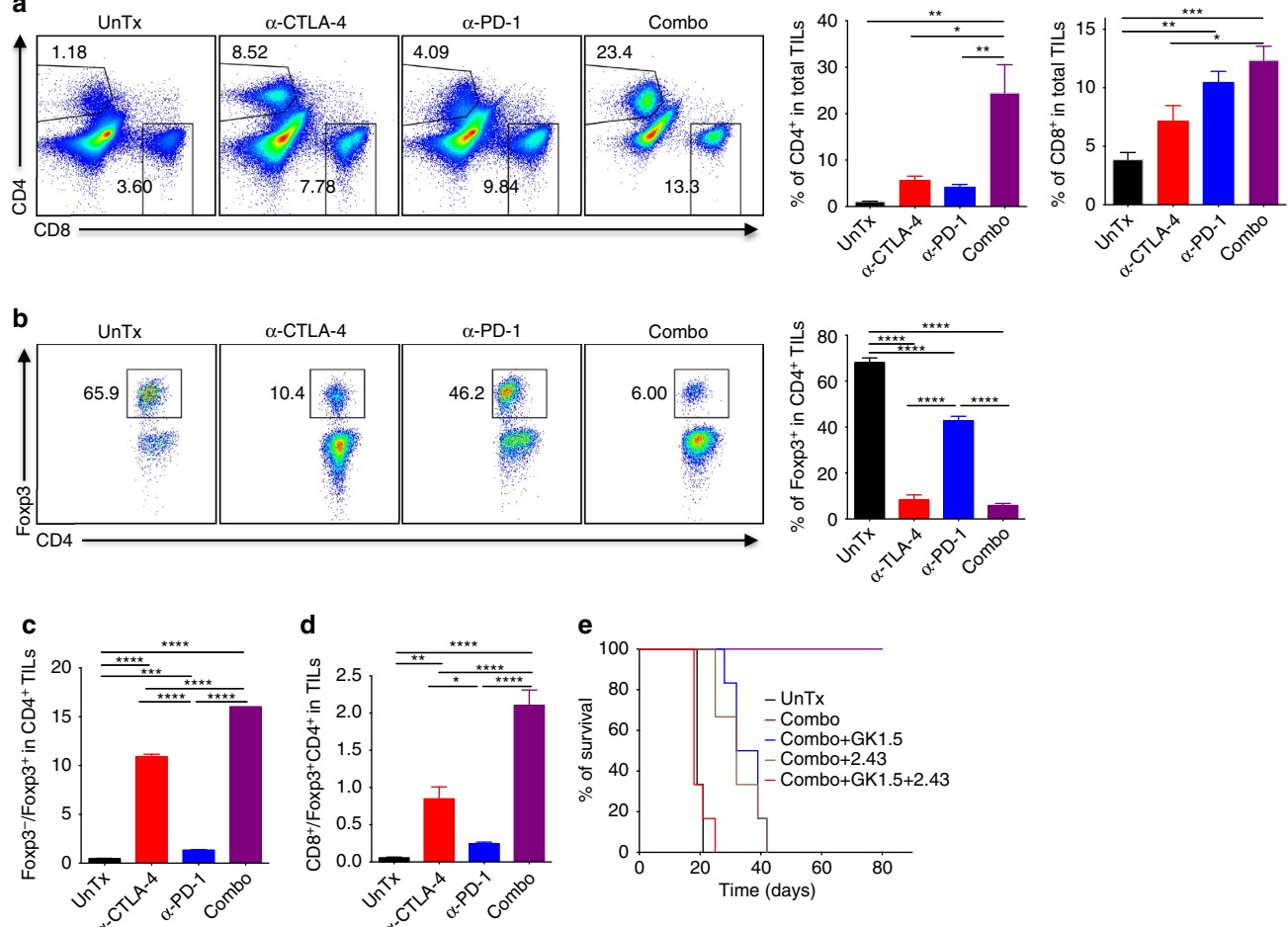

**Figure 3 | Combination therapy-mediated MB49 tumour rejection is dependent on CD4$^+$ and CD8$^+$ T cells.** Isolated TILs from 13–15 days tumours, 1–3 days after the last treatment with α-CTLA-4, α-PD-1, combination of both (Combo) were analyzed by flow cytometry for (**a**) Total CD4$^+$ T cells and CD8$^+$ T cells. Representative flow panels are shown on the left and pooled results from five mice are depicted in bar graphs on the right. (**b**) Foxp3$^+$ $T_{reg}$ in CD4$^+$ T cells. Pooled results from five mice are shown in the right bar graph. (**c**) Ratios of Foxp3$^-$ to Foxp3$^+$ in CD4$^+$ T cells. (**d**) Ratios of CD8$^+$ T cells to Foxp3$^+$ CD4$^+$ T cells. Mice depleted of CD4$^+$ (GK1.5), CD8$^+$ (2.43) or both (GK1.5 + 2.43) were treated with the combo and mouse survivals are presented (**e**). Data in the bar graphs are means ± s.e.m. *$P < 0.05$; **$P < 0.01$; ***$P < 0.001$; ****$P < 0.0001$ by one-way ANOVA with Bonferroni's *post hoc* test. Data are representative of three independent experiments.

Our findings corroborate a recent report that a correlation between central memory phenotype in CD8$^+$ cells and rejection of metastatic osteosarcoma K7M2 by combined α-CTLA-4 and α-PD-L1 (ref. 11).

Considering the pivotal role of IL-7/IL-7Rα signalling pathway in the generation and maintenance of memory T cells, in particular the long-lived $T_{CM}$[34–36], we ask if combination therapy engages IL-7-IL-7Rα signalling in TILs by analyzing IL-7Rα expression on CD4$^+$ and CD8$^+$ TILs (Fig. 5d) treated with combination therapy. Clearly, combination therapy upregulated IL-7Rα, which likely renders TILs more advantageous in competing for the limited amount of IL-7 (ref. 36). Considering augmented IL-7Rα expression on TILs and increased IL-2$^+$ IFN-γ$^+$ dual producers by combination therapy (Fig. 4b; Supplementary Fig. 3A–C), we reasoned that combination therapy enables T cells to better survive nutrient- and/or cytokine-depleted TME. Upregulation of IL-7Rα by combination therapy was further confirmed by real-time RT-PCR analysis of sorted CD4$^+$ and CD8$^+$ TILs with IL-7Rα specific primers (Fig. 5e).

Next, we explored if upregulation of IL-7Rα is functionally important for combination therapy-induced tumour rejection by inoculating $Il7r^{-/-}$ mice, deficient of IL-7Rα with MB49 tumour

cells, followed by combination therapy. In contrast to the 100% protection in WT mice, combination therapy failed to cure $Il7r^{-/-}$ tumour-bearing mice, which rapidly died within 20 days post-tumour inoculation (Fig. 5f), reflecting a critical role of IL-7-IL-7Rα signalling in combination therapy. To circumvent chronic immune defects in $Il7r^{-/-}$ mice, we blocked IL-7 signalling in MB49 tumour-bearing mice before combination therapy. This short-term blockade resulted in a partial yet significant attenuation of therapeutic effects of combination therapy (Fig. 5g). This partial response could be due to incomplete blockade of IL-7 signalling with the current treatment regimen. Nevertheless, these two complementary approaches strongly support that the IL-7/IL-7Rα signalling pathway is required for optimal therapeutic effects of combination therapy.

We also attempted to assess if IL-7R signalling is involved in combination therapy-induced T-cell infiltration and effector function of TILs. However, while T cells can be readily detected in the spleens of $Il7r^{-/-}$ tumour-bearing mice (Supplementary Fig. 4A), we detected very few T cells in the tumours of $Il7r^{-/-}$ mice (Fig. 5h), suggesting a role for IL-7R signalling in T-cell trafficking into tumour and/or survival of T cells in tumour. This paucity of TILs in $Il7r^{-/-}$ mice prohibited further

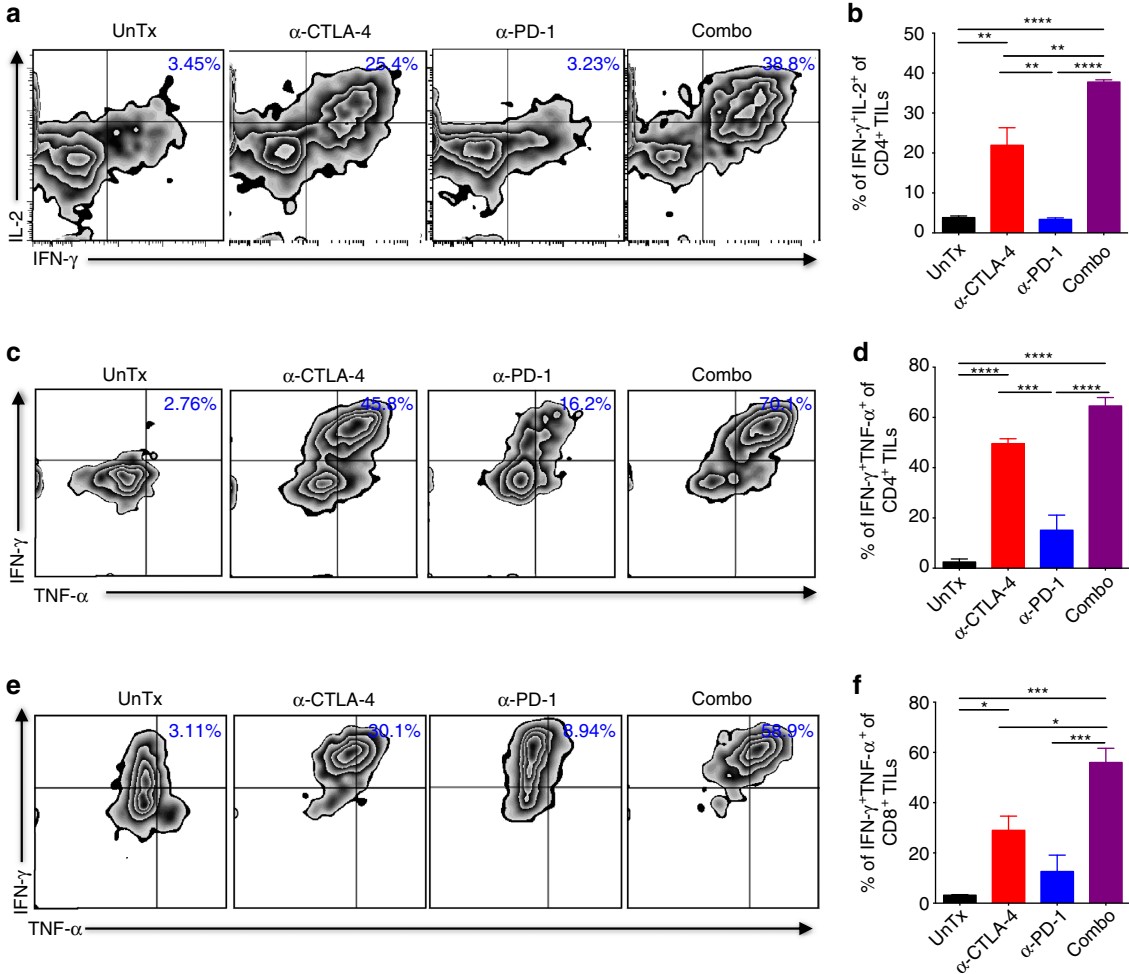

**Figure 4 | Combination therapy enhances polyfunctionality of CD4$^+$ and CD8$^+$ TILs. (a)** Purified CD4$^+$ from TILs described as in Fig. 3 co-cultured with irradiated MB49 cells and splenic DCs *in vitro* were analyzed to determine the percentages of cells positive for IL-2 and IFN-γ. **(b)** Summarized results of dual IFN-γ$^+$IL-2$^+$ producers from five mice. **(c–f)** Isolated TILs without further purification were stimulated briefly with PMA and ionomycin *in vitro*, and then analyzed for IFN-γ and TNF-α production in CD4$^+$ TILs **(c)** or in CD8$^+$ TILs **(e)**. Percentages of dual IFN-γ$^+$TNF-α$^+$ producers in CD4$^+$ or CD8$^+$ TILs from fiev mice are depicted in **(d)** and **(f)**, respectively. Data in the bar graphs are means ± s.e.m. *$P < 0.05$; **$P < 0.01$; ***$P < 0.001$; ****$P < 0.0001$ by one-way ANOVA with Bonferroni's *post hoc* test. Data are representative of three independent experiments.

functional analysis. In addition, we observed that combination therapy greatly reduced abundance of both CD4$^+$ and CD8$^+$ T cells in the spleen of *Il7r*$^{-/-}$ mice (Supplementary Fig. 4A), which was not seen in WT (Supplementary Fig. 4B), indicating IL-7R signalling is instrumental for survival of peripheral T-cell upon combination treatment. Taken together, combination therapy elicits strong protective immunity against re-challenge and activates IL-7R signalling in TILs, which is instrumental in regulating antitumour effects of combination therapy.

**Combination therapy requires both IL-7 and IFN-γ signalling**. Another well-recognized important pathway in anti-tumour immune response is the IFN-γ signalling pathway[30]. A recent paper using human T cells isolated from patients treated with α-CTLA-4, α-PD-1 or combination therapy identified IFN-γ as the only commonly-upregulated gene, suggesting a prominent role of IFN-γ signalling in immune checkpoint therapy[37], consistent with our observation that combination therapy markedly increased IFN-γ production by TILs. However, detailed mechanistic studies on how IFN-γ signalling pathway regulates therapeutic effects of immune checkpoint therapy are lacking. To address this, we challenged WT or *Ifngr1*$^{-/-}$ mice, which are deficient of IFN-γR1 (the essential receptor for IFN-γ)

with MB49 tumour cells, followed by combination treatment. While combination therapy protected WT mice from MB49 tumour challenge, this protective effect was abolished in *Ifngr1*$^{-/-}$ mice (Fig. 6a). Utilizing a neutralizing antibody (XMG1.2) to block IFN-γ signalling, we also observed a complete abrogation of combination therapy-induced MB49 tumour rejection (Fig. 6b). Collectively, these results provide direct evidence that IFN-γ signalling in host cells governs combination therapy-induced tumour eradication.

We next asked if the IFN-γ signalling pathway controls therapeutic effects of combination therapy by regulating the enhanced immune responses as described above. Convincingly, absence of IFN-γR1 in host cells diminished immune-boosting effects of combination therapy, including increased T-cell infiltration into tumours, depletion of intratumoral $T_{reg}$, augmented effector/$T_{reg}$ ratios of TILs, enhanced polyfunctionality of CD4$^+$ and CD8$^+$ TILs, and elevated Ki-67$^+$IFN-γ$^+$ CD4$^+$ and CD8$^+$ TILs (Supplementary Fig. 5). We therefore concluded that IFN-γ signalling in the host cells has a crucial role in mediating therapeutic effects of combination therapy likely by modulating T-cell-mediated responses.

Considering both IL-7 and IFN-γ signalling in host cells are required for combination therapy-induced tumour rejection, a

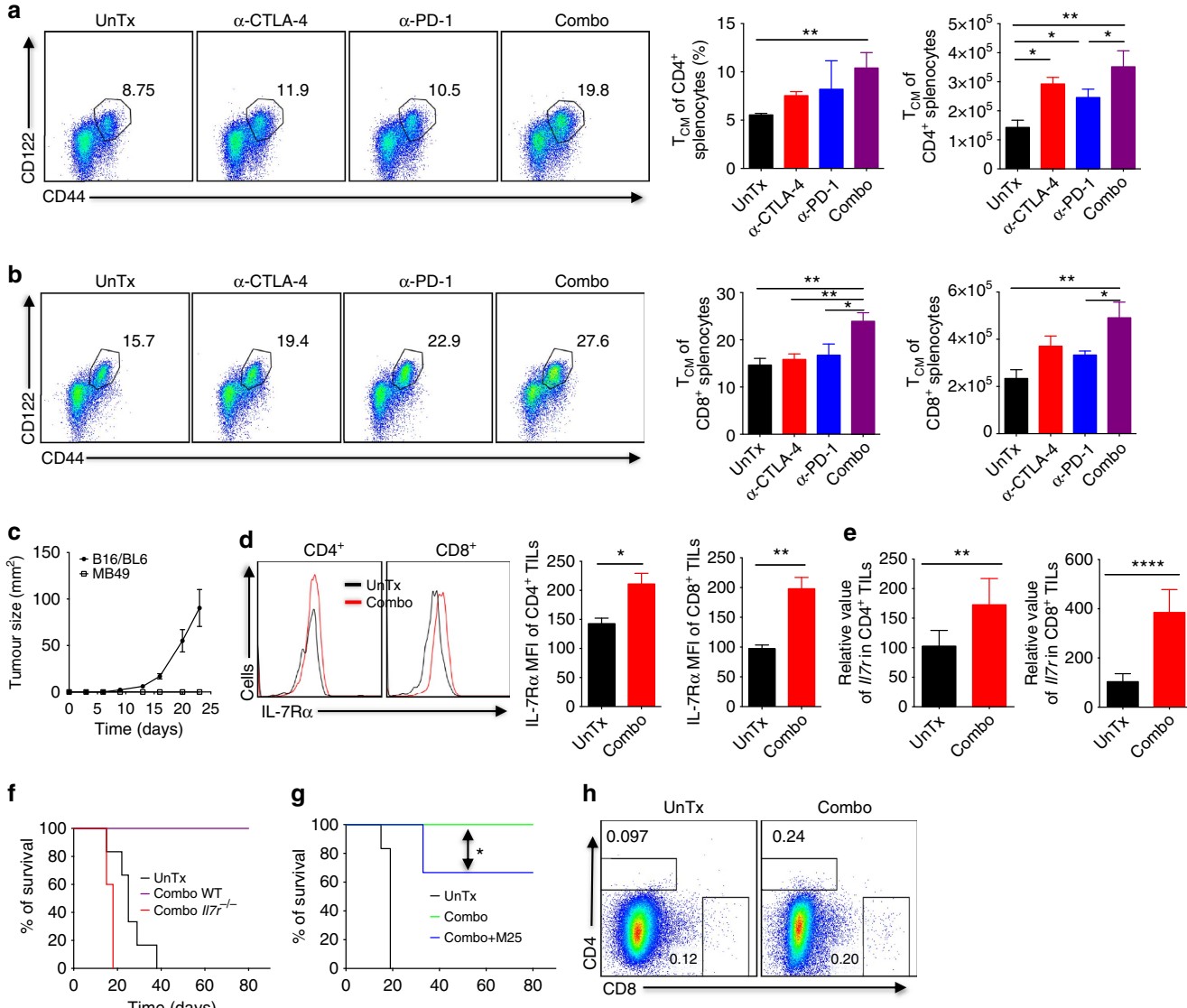

**Figure 5 | Combination therapy induces memory response and therapeutic benefits by engaging IL-7 signalling.** (**a**,**b**) Isolated splenocytes from tumour-bearing WT mice with regressing tumours treated with mono- or combo therapy were analyzed for: (**a**) Frequencies of CD4$^+$CD44$^{hi}$CD122$^+$ ($T_{CM}$) and (**b**) Frequencies of CD8$^+$CD44$^{hi}$CD122$^+$ ($T_{CM}$) cells. Pooled results of percentage and absolute cell number of $T_{CM}$ from five mice are depicted as bar graphs. (**c**) Combo-treated WT mice that previously rejected MB49 tumours were re-challenged with MB49 tumour cells or unrelated B16-BL6 melanoma cells. Tumour growth (mm$^2$) is shown. (**d**) Surface expression of IL-7Rα on CD4$^+$ and CD8$^+$ TILs as described in Fig. 3 and MFIs of IL-7Rα from five mice are presented (right). (**e**) mRNA expression of IL-7Rα in sorted CD4$^+$ and CD8$^+$ TILs from untreated and combo-treated mice by real-time RT-PCR. Results were normalized to the expression of housekeeping gene ($\beta$-actin). (**f**) Survival of WT and $Il7r^{-/-}$ tumour-bearing mice treated with combination therapy. (**g**) Survival of tumour-bearing mice pretreated with blocking antibody against IL-7 (M25) upon combination therapy. (**h**) Abundance of CD4$^+$ and CD8$^+$ TILs isolated from $Il7r^{-/-}$ tumour-bearing mice as described in **f**. Data are means ± s.e.m. of five mice in each group. NS, no statistical significance; *$P<0.05$; **$P<0.01$; ***$P<0.001$; ****$P<0.0001$ by one-way ANOVA with Bonferroni's *post hoc* test (**a**,**b**), two-tailed unpaired Student's *t*-test (**d**,**e**) or log-rank (Mantel–Cox) test (**g**). Data are representative of two to three independent experiments.

process dependent on T cells (Fig. 3e), we speculated that IL-7 and IFN-γ signalling in T cells dictate the immunotherapeutic benefits of combination therapy. To test this, we adoptively transferred purified WT, $Ifngr1^{-/-}$ and $Il7r^{-/-}$ total T cells into sublethally irradiated CD45.1 mice, followed by tumour inoculation and combination therapy. As shown in Fig. 6c, although tumour growth appeared to be comparable among all cohorts in the early stage, combination therapy significantly reduced tumour volume in mice that were adoptively transferred with WT total T cells, but not in mice that received either $Il7r^{-/-}$ or $Ifngr1^{-/-}$ total T cells, pointing to an indispensable role of IL-7r and IFN-γ signalling in T cells in this process.

Furthermore, we adoptively transferred total WT, $Ifngr1^{-/-}$ or $Il7r^{-/-}$ T cells into Rag-1$^{-/-}$ mice that lack mature T and B cells but contain innate immune cells such as DCs and macrophages with intact IL-7 and IFN-γ signalling. Thus, the only T cells present in these mice are those transferred, which allow us to unequivocally evaluate if IL-7 and IFN-γ signalling in T cells is critical in mediating combination therapy. We observed similar outcomes to those using CD45.1 mice as the recipients (Fig. 6d), further supporting the predominant function of IFN-γ and IL-7 signalling in T cells but not other immune cells in driving antitumour effects of combination therapy. To assess if combination therapy-improved mouse survival also depends on

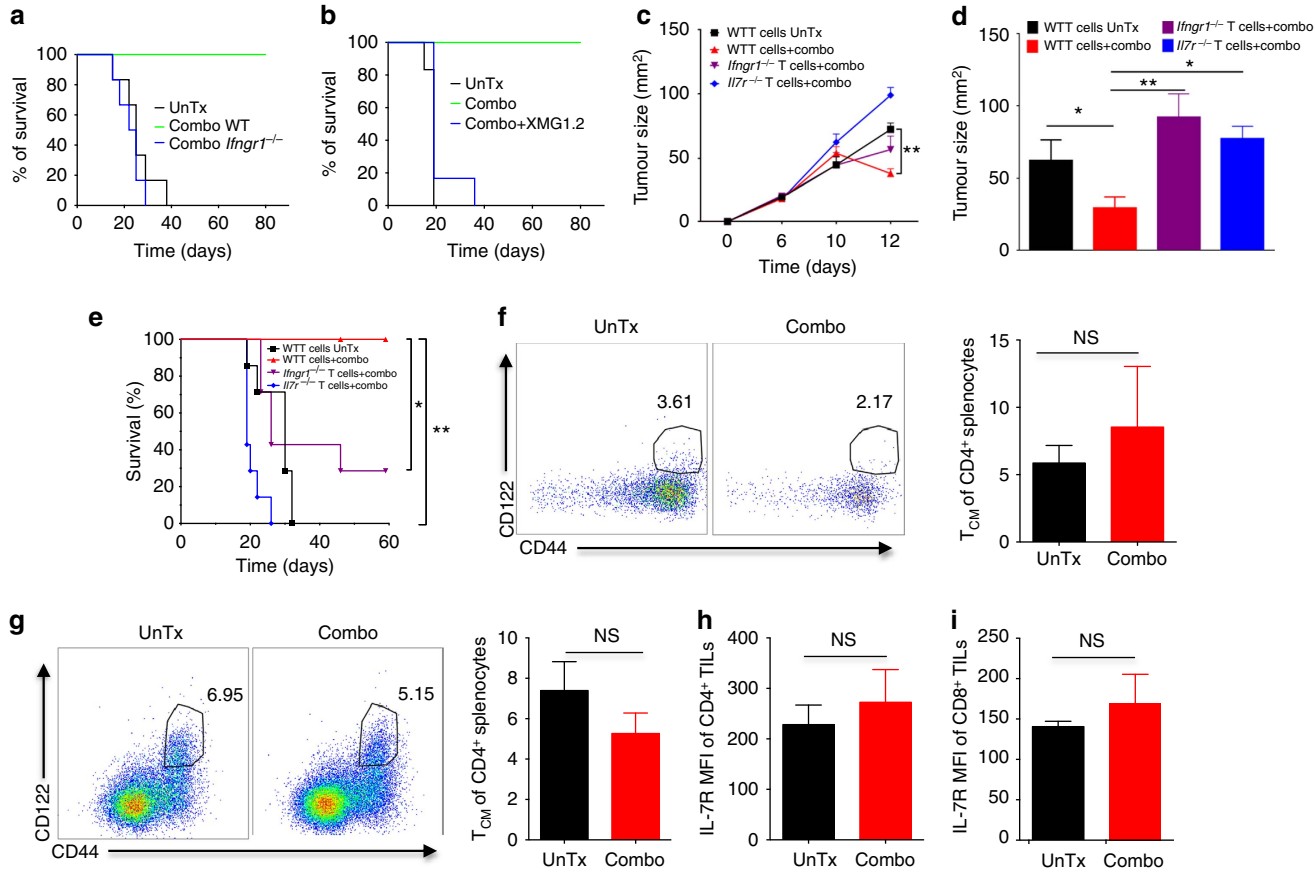

**Figure 6 | Combination therapy requires an intact loop of IL-7 and IFN-γ signalling pathways to reject MB49 tumours.** (**a**) Survival of WT and *Ifngr1*$^{-/-}$ MB49 tumour-bearing mice untreated (UnTx) or treated with combo. (**b**) Survival of tumour-bearing mice pretreated with blocking antibody against IFN-γ (XMG1.2) and then treated with combination therapy. (**c**) Sublethally irradiated CD45.1 mice were adoptively transferred with WT, *Ifngr1*$^{-/-}$ or *Il7r*$^{-/-}$ total T cells, followed by MB49 tumour inoculation and combination therapy. Tumour growth kinetics is shown. (**d**) Rag-1$^{-/-}$ mice were adoptively transferred with WT, *Ifngr1*$^{-/-}$ or *Il7r*$^{-/-}$ total T cells, followed by MB49 tumour inoculation and combination therapy. Tumour sizes on day 12 post-tumour inoculation are shown. (**e**) CD45.1 mice were treated as described above in **c** and mouse survivals are illustrated. (**f,g**) Isolated splenocytes from *Il7r*$^{-/-}$ (**f**) and *Ifngr1*$^{-/-}$ (**g**) tumour-bearing mice as described in Fig. 3 were analyzed for frequencies of CD4$^{+}$CD44$^{hi}$CD122$^{+}$ ($T_{CM}$) cells. Pooled results from five mice are depicted as bar graphs. (**h,i**) Isolated TILs from *Ifngr1*$^{-/-}$ tumour-bearing mice treated with combination therapy were examined for IL-7Rα expression in CD4$^{+}$ (H) or CD8$^{+}$ (**i**). Data are means ± s.e.m. of five mice in each group. NS, no statistical significance; *$P < 0.01$; **$P < 0.01$ by one-way ANOVA with Bonferroni's *post hoc* test (**c,d**), log-rank (Mantel–Cox) test (**e**) or two-tailed unpaired Student's *t*-test (**f-i**). Data are representative of two to three independent experiments.

IL-7 and IFN-γ signalling in T cells, we followed long-term survival of CD45.1 tumour-bearing mice that were infused with WT, *Ifngr1*$^{-/-}$ or *Il7r*$^{-/-}$ total T cells. As depicted in Fig. 6e, combination therapy-improved survival of tumour-bearing mice that received WT total T cells but remained largely ineffective in those with adoptively transferred *Il7r*$^{-/-}$ or *Ifngr1*$^{-/-}$ total T cells. In sum, both suppression of tumour growth and improved mouse survival by combination therapy require intact IL-7 and IFN-γ signalling in T cells.

To shed light on how IL-7 and IFN-γ signalling in T cells affect T-cell proliferation, infiltration into tumours and functionality using our adoptive transfer system, we labelled purified total WT, *Il7r*$^{-/-}$ or *Ifngr1*$^{-/-}$ T cells (congenically marked by CD45.2) with CFSE before the adoptive transfer into CD45.1 mice, followed by MB49 tumour inoculation and combination therapy. Lack of IFN-γR1 or IL-7Rα in CD4 T cells and CD8 T cells appeared not to impact their proliferative ability, evidenced by comparable dilution of CFSE to that of WT T cells (Supplementary Fig. 6A,B). However, there was a significant reduction of *Ifngr1*$^{-/-}$ or *Il7r*$^{-/-}$ T-cell infiltration into tumours, with the latter being more severe (Supplementary Fig. 6C,D), which prevented further analysis of how IL-7Rα

deficiency affects intratumoral T-cell effector function and $T_{reg}$ abundance. Additional analysis of tumour-infiltrating *Ifngr1*$^{-/-}$ T cells revealed a significantly higher proportion of $T_{reg}$ in CD4$^{+}$ T cells, reduced ratios of CD4$^{+}$ non-$T_{reg}$/$T_{reg}$ and CD8$^{+}$/$T_{reg}$ cells, and decreased CD4$^{+}$IFN-γ$^{+}$ TILs, as compared to tumour-infiltrating WT T cells (Supplementary Fig. 6E–G). Taken together, these results corroborate our above findings in KO mice that IL-7 and IFN-γ signalling in T cells modulate immune responses and tumour inhibition by combination therapy.

Since both *Ifngr1*$^{-/-}$ and *Il7r*$^{-/-}$ mice are incapable of eliminating primary tumours upon combination therapy, we contemplated memory cell expansion is defective in these KO mice. To this end, we analyzed $T_{CM}$ cells in the spleen of *Il7r*$^{-/-}$ and *Ifngr1*$^{-/-}$ tumour-bearing mice treated with or without combination therapy. In contrast to the induced expansion of CD4$^{+}$ and CD8$^{+}$ $T_{CM}$ populations by combination therapy in WT mice (Fig. 5a,b), no overt expansion of CD4$^{+}$ $T_{CM}$ was observed in *Il7r*$^{-/-}$ (Fig. 6f) or *Ifngr1*$^{-/-}$ (Fig. 6g) mice. Similarly, no obvious expansion of CD8$^{+}$ $T_{CM}$ was found in *Il7r*$^{-/-}$ or *Ifngr1*$^{-/-}$ mice (Supplementary Fig. 7A,B). These results demonstrated that both IL-7R and IFN-γR signalling

pathways have important roles in regulating endogenous memory response upon combination therapy. This similar yet non-redundant function of IFN-γ and IL-7 signalling prompted us to explore if IFN-γ and IL-7 signalling pathways mutually regulate each other in TILs. While we could not examine how IL-7Rα deficiency affects IFN-γR1 expression in TILs due to rare TILs in $Il7r^{-/-}$ mice, we found deficiency of IFN-γR1 abrogated IL-7Rα upregulation on CD4$^+$ (Fig. 6h) and CD8$^+$ (Fig. 6i) TILs that was induced on WT TILs by the combination therapy (Fig. 5d), suggesting IFN-γ signalling in stromal cells modulates IL-7Rα expression in TILs. Taken together, we show combination therapy induces expansion of $T_{CM}$ cells and elicits protective immunity against subsequent re-challenge, which are dependent on both IFN-γ and IL-7 signalling in T cells. More importantly, we identify an intricate interplay of IFN-γ and IL-7 signalling in TILs that dictates anti-tumour immune responses and therapeutic effects of combination therapy (Supplementary Fig. 7C).

## Discussion

The recent FDA approvals of α-CTLA-4 and α-PD-1 have opened a new era in the treatment of cancer termed 'immune checkpoint blockade'. The impressive durability but low response rate achieved with these treatments requires novel strategies to improve response rate, among which combination therapy of α-CTLA-4 and α-PD-1 has attracted intense interests. While convincing outcomes with combination therapy have been reported in patients with different types of cancers[12,14,15], the underlying mechanisms remain to be illustrated. Here, we provide direct evidence that PD-1 is a primary suppressive pathway utilized by bladder tumour cells to 'escape' α-CTLA-4 monotherapy by showing that α-CTLA-4 therapy upregulates PD-1/PD-L1 expression and that bladder tumour cells express high levels of PD-L1. As a result, concomitant blocking of both α-PD-1 and α-CTLA-4 led to complete rejection of bladder tumours, in contrast to the inefficient monotherapies, an outcome further confirmed in the Renca model for renal cancer, which parallels preliminary results from a phase I clinical trial of combination therapy in metastatic RCC patients[14,38]. Mechanistically, we demonstrate that combination therapy synergistically or additively induced greater immune responses including T-cell infiltration into tumours, depletion of intratumoral $T_{reg}$, and induction of polyfunctional effector TILs. *In vivo* depletion with neutralizing antibodies revealed that CD4$^+$ and CD8$^+$ T cells but not NK and NKT cells are essential players in combination therapy-mediated tumour rejection. Since GK1.5 antibody depletes total CD4$^+$ T cells, including $T_{eff}$ as well as $T_{reg}$, we could not distinguish their respective contributions to therapeutic benefits of combination therapy. At the molecular level, using genetic $Ifngr1^{-/-}$ and $Il7r^{-/-}$ mice and different adoptive transfer systems with WT, $Ifngr1^{-/-}$ and $Il7r^{-/-}$ total T cells, we illustrate that the interdependence between IL-7 and IFN-γ signalling in T cells underpins the anti-tumour immune responses elicited by the combination therapy.

A cardinal feature of immunogenic tumour rejection by immune checkpoint therapy is the induced durable memory response. Indeed, we found combination therapy led to expansion of memory T cells and protected mice from re-challenge, associated with upregulation of IL-7Rα on CD4$^+$ and CD8$^+$ TILs. Since IL-7 is produced at a relatively fixed amount[36], upregulation of IL-7Rα in intratumoral T cells could be instrumental for their survival in the hostile TME so as to better execute their anti-tumour activity. The importance of IL-7 signalling in combination therapy is demonstrated by our data that either IL-7Rα deficiency or IL-7 blockade significantly abolished therapeutic benefits of combination treatment.

Theoretically, targeting IL-7 signalling should hold greater promises than targeting IL-2 signalling, due to the selective expansion of effector and memory T-cell subsets by IL-7 without expanding $T_{reg}$ (ref. 39), in contrast to undifferentiated expansion of both effector T cells and $T_{reg}$ by IL-2. Recently, it was reported that CAR-engrafted Espstein–Barr-virus-specific CTLs overexpressing IL-7Rα (ref. 39) has yielded some successes in preclinical studies. Along this line, preclinical application of IL-7 yields substantial antitumour activity in different murine tumour models by expanding effector CD4$^+$ and CD8$^+$ T cells[40–42]. However, two phase I clinical trials of IL-7 in patients with cancer failed to generate overt anti-tumour effects, either as a standalone treatment[43] or in combination with vaccination[44], despite the expected expansion of CD4$^+$ and CD8$^+$ T cells in IL-7 treated patients[43]. Given the dual roles of IL-7 in promoting $T_{eff}$ survival and expansion[40,43] as well as in suppressing $T_{reg}$ function [41], further explorations of combined immune checkpoint blockade and IL-7 supplementation could potentially improve antitumour therapy, as our data suggested. On a side note, identification of biomarkers that can predict which patients respond to cancer immunotherapy is an active research area[45]. On the basis of our results and the reported requirement of IL-7Rα expression on effector T cells for their survival upon transfer[46], we predict that upregulation of IL-7Rα could represent a potential biomarker, which warrants further analysis of IL-7Rα expression in responders versus non-responders treated with immune checkpoint therapy. In addition to IL-2 and IL-7, other cytokines such as IL-4, IL-9, IL-15 and IL-21 share the common γ-chain receptor to emanate their downstream signals. Thus, it will be interesting to explore if these cytokine signalling pathways also have a role in this process.

Despite the recognized involvement of IFN-γ in anti-tumour response, detailed analysis of $Ifngr1^{-/-}$ mice with defective IFN-γ signalling in the context of tumour immunotherapy has been lacking. Here, we demonstrate that IFN-γ signalling in host T cells controls therapeutic effects of combination therapy by modulating T-cell-mediated immune responses. Recently, Schreiber and colleagues[32] reported that α-PD-1 alone impacted mostly pathways associated with metabolism, α-CTLA-4 alone affected pathways associated with cell cycle and effector memory, and the combination altered T-cell effector pathway. Thus, it will be interesting to evaluate how IFN-γ/IFN-γR1 axis affects these pathways in the context of α-CTLA-4 and/or α-PD-1 treatment in the future. It might also be worthwhile to explore how IFN-γ/IFN-γR1 axis interacts with FcγRIV signalling to drive intratumoral depletion of $T_{reg}$ upon immune checkpoint blockade, as our current data and previous reports suggested both pathways are required in this process[47,48].

On the basis of our findings of abrogation of therapeutic effects of combination therapy by T-cell depletion, correlation of diminished therapeutic efficacy of combination therapy to attenuated T-cell-mediated immune responses in $Ifngr1^{-/-}$ and $Il7r^{-/-}$ mice, loss of combination therapy-induced IL-7Rα upregulation in $Ifngr1^{-/-}$ TILs, and inability of adoptively transferred $Ifngr1^{-/-}$ or $Il7r^{-/-}$ total T cells to suppress tumour growth and to improve mouse survival upon combination therapy, we propose that an intricate network consisting of interdependent IFN-γ and IL-7 signalling pathways (not exclusively) in T cells controls anti-tumour activity of combination therapy. Consistent with our finding, the interplay between IFN-γ and IL-7 signalling pathways has been previously demonstrated in other experimental systems. In human intestinal epithelial cells, IFN-γ can induce IL-7 gene expression via binding of its downstream target IFN regulatory factor-1 (IRF-1) to an IFN-stimulated response element (ISRE) located in the 5′ upstream region of the IL-7 gene[49]. Along this line, deficiency

of IFN-γR1 completely diminishes the induction of atypical IL-7Rα[+] progenitors in bone marrow by acute malaria infection, indicating an essential role of IFN-γ signalling in mediating IL-7Rα on bone marrow cells[50]. Further, a recent report demonstrated that IL-12 conditioning, potentially by inducing IFN-γ production, led to increased IL-7Rα expression on polarizing Tc1 effector cells, associated with enhanced *in vivo* persistence and antitumour efficacy of CD8[+] T cells upon adoptive transfer[46]. Vice versa, an early report suggests that IL-7R signalling contributes to the maintenance of IFN-γ-producing Th1 effector cells during chronic *L. major* infection[51]. Thus, our finding corroborates previous reports and reveals for the first time an interactive loop of IFN-γ and IL-7 signalling pathways in TILs that dictates antitumour activity of combination therapy (Supplementary Fig. 7C). It remains to be tested if this interdependent loop between IFN-γ and IL-7 signalling pathways can be extended to human tumours. In our future studies, we propose to analyze IL-7Rα expression in human tumours treated with immune checkpoint therapy. Given the considerable similarities between human bladder tumours and murine MB49 bladder tumours, a recent report of IFN-γ upregulation in patients treated with immune checkpoint blockade[37], and the essential role of IL-7 and IFN-γ sigaling in T-cell survival and function, we reason such an interdependent loop likely exsists to mediate beneficial effects of combination therapy in human tumours.

Our results are different from a previous report[52] using MB49 tumour model where the authors observed comparable therapeutic efficacy between α-CTLA-4 monotherapy and combination therapy. This discrepancy is likely due to the fact that female mice were used in that study and thus the mismatch gender effects on immune responses may have created strong anti-tumour immunity that prevented interpretation of the impact of immune checkpoint blockade treatment. Considering our original finding of a partial response of patients with bladder cancer to α-CTLA-4 monotherapy and the recent report of a noteworthy activity of α-PD-L1 antibody MDPDL3280A in patients with bladder cancer[53], we reason that combination therapy represents a more effective approach, which is consistent with our observations of the high expression of PD-L1 on MB49 cells, upregulation of PD-1 induced by α-CTLA-4 treatment in both human and murine bladder tumour samples, and strong synergism of α-CTLA-4 and α-PD-1 in tumour rejection.

In summary, we demonstrated that combination therapy of α-CTLA-4 and α-PD-1 elicited stronger anti-tumour immune responses, potent MB49 tumour rejection, and induced protective immunity against re-challenge. Combination therapy-mediated tumour eradication requires presence of T cells and an intact loop of IFN-γ and IL-7 signalling in T cells. Our findings suggest a potential combination therapy of immune checkpoint therapy and IL-7 signalling to boost efficacy.

## Methods

**Mice, cell lines and reagents.** Male C57BL/6 (B6, 5–7 weeks) mice were purchased from the National Cancer Institute (Frederick, MD). CD45.1, Rag-1[−/−], *Il7r*[−/−] and *Ifngr1*[−/−] mice, all on the B6 background, were obtained from The Jackson Laboratory (Bar Harbor, ME, USA). Balb/c female mice were also from The Jackson Laboratory. All mice were kept in specific pathogen-free conditions in the Animal Resource Center at The University of Texas MD Anderson Cancer Center (MDACC). Animal protocols were approved by the Institutional Animal Care and Use Committee of MDACC.

The chemically-induced murine bladder carcinoma MB49 cell line, kindly provided by Dr A. Kamat at MD Anderson Cancer Center, was originated from a male C57BL/6 mouse[21] and cultured in Dulbecco's modified Eagle's medium with 10% fetal bovine serum (FBS) at 37 °C, 5% CO₂. The B16/BL6 murine melanoma cells, originally obtained from Dr I. Fidler at MD Anderson Cancer Center, were grown in MEM supplemented with 10% FBS, 2 mM L-glutamine, 1 mM sodium pyruvate, 1% non-essential amino acids and vitamin (all from Invitrogen).

Murine renal adenocarcinoma cell line RENCA was purchased from ATCC and cultured in RPMI-1640 medium supplemented with 10% FBS, non-essential amino acids (0.1 mM extra) and additional L-glutamine (2 mM extra). All cell lines are mycoplasma-free.

α-CTLA-4 (clone 9H10) and α-PD-1 (RMP1-14) blockade antibodies, as well as depleting antibodies against CD4 (GK1.5), CD8 (2.43), NK1.1 (PK136), IFN-γ (XMG1.2) and IL-7 (M25) were all purchased from BioXcell (West Lebanon, NH, USA).

***In vivo* tumour inoculation and treatment.** C57BL/6 male and Balb/c female mice (5 or 10 mice pergroup) were inoculated subcutaneously in the right flank with $5 \times 10^4$ MB49 and $2 \times 10^5$ RENCA cells, respectively, on day 0. On day 6 post-MB49 inoculation or day 8 post-RENCA inoculation when tumours became palpable, mice were injected intraperitoneally (i.p.) α-CTLA-4, α-PD-1 or both. The first dose was 200 μg per mouse and the treatments were repeated twice, every 3 days with a dose of 100 μg per mouse. For adoptive transfer of total T cells, sublethally irradiated CD45.1 mice (600 Rads) or Rag-1[−/−] mice were injected via tail vein 5–10 million of purified total T cells isolated from WT, *Il7r*[−/−] or *Ifngr1*[−/−] mice, followed by tumour inoculation of MB49 cells on the next day and subsequent combination treatments as described above. For some experiments, purified WT, *Il7r*[−/−] or *Ifngr1*[−/−] total T cells were labelled with CFSE, before adoptive transfer. Tumour growth measurements were taken 2–3 times per week, starting from day 6 after tumour inoculation. Mice were euthanized when tumour reached 1.5 cm in diameter or ulceration or moribund occurred, recorded as death for the survival curve.

To deplete CD4[+], CD8[+] or NK1.1[+] cells, 250 μg of anti-CD4 (GK1.5), anti-CD8 (2.43), anti-NK1.1 (PK136) or control antibodies in 200 μl of D-PBS were injected i.p. into each mouse on the day before tumour challenge, followed by three injections on days 1, 3 and 10 after tumour injection. Blockade of IFN-γ or IL-7 was performed by the i.p. injection of XMG1.2 or M25 antibody (250 μg/200 μl/mouse) before tumour challenge and once every 3 days after tumour challenge.

For the re-challenge experiments, combination therapy-treated mice free of tumour were injected subcutaneously with $5 \times 10^5$ MB49 or $2 \times 10^4$ B16-BL6 melanoma cells. Tumour growth was monitored as described above. Tumour size was calculated as length × width (mm²).

**Flow cytometric analysis.** For analysis of surface markers, cells were stained in PBS containing 2% (wt/vol) BSA, with anti-CD4-PB (RM4-5) (Biolegend, 100531, 1:200), anti-CD8α-BV786 (53-6.7) (BD Horizon, 563332, 1:200), anti-TCRβ-APC Cy7 (H57-597) (BD Pharmingen, 560656, 1:200), anti-CD45.2-BV605 (104) (Biolegend, 109841, 1:200), anti-PD-1-APC (RMP1-30) (eBiosciences, 17-9981-82, 1:200), anti-PD-L1-BV711 (10F.9G2) (Biolegend, 124319, 1:100) and anti-CTLA-4-PE (UC10-4B9) (eBiosciences, 12-1522-82, 1:200) on ice for 30 min. Intracellular anti-Foxp3-AF532 (FJK-16 s) (eBiosciences, 58-5773-82, 1:100), anti-IL-2-PerCp Cy5.5 (PC61.5) (eBiosciences, 45-0251-82, 1:100), anti-Ki-67-PE-Cy7 (SolA15) (eBiosciences, 25-5698-82, 1:300), anti-IFN-γ-FITC (XMG1.2) (eBiosciences, 11-7311-82, 1:200) and anti-CTLA-4-PE (UC10-4B9) (eBiosciences, 12-1522-82, 1:200), anti-TNF-α-BV650 (MP6-XT22) (Biolegend, 506333, 1:100) were analyzed by flow cytometry according to the manufacturer's instructions. Flow cytometry data were acquired on LSRII or BD FACS Canton II (BD Biosciences) and analyzed using Flowjo software (Tree Star).

**TILs isolation and cytokine analysis.** TILs and splenocytes isolated from tumour-bearing mice were analyzed accordingly. In some experiments, CD4[+] TILs were further purified using Dynabeads FlowComp (Invitrogen) (>85% purity). The purified $1 \times 10^5$ CD4[+] T cells were co-cultured with $5 \times 10^4$ splenic DCs plus irradiated $1 \times 10^6$ MB49 tumour cells (150 Gy) or antigen-unrelated B16/BL6 melanoma cells for 18 h with monensin added for the last 2 h. Alternatively, TILs were stimulated for 4–5 h with PMA plus ionomycin in the presence of monensin. Stimulated cells were washed once with FACS buffer (D-PBS + 2% FBS) before being stained according to the manufacturer's instructions (eBioscience).

**RNA analysis.** RNA was extracted from sorted CD4[+] and CD8[+] TILs from untreated or combination therapy-treated mice using an RNeasy kit (Qiagen). cDNA was synthesized with SuperScript III reverse transcriptase (Invitrogen). An ABI 7500 Real-time PCR system was used for quantitative RT-PCR, with *Il7r* primers purchased from Origen (Forward, 5′-CACAGCCAGTTGGAAGTGG ATG-3′; reverse, 5′-GGCATTTCACTCGTAAAAGAGCC-3′). The cycling threshold value of the endogenous control gene (β-actin) was subtracted from the cycling threshold value of *Il7r* to generate the change in cycling threshold ($\Delta C_T$). The expression of *Il7r* is presented as the 'fold change' relative to that of untreated TILs ($2^{-\Delta\Delta CT}$), as described[54].

**Immunohistochemistry (IHC).** Informed consent was obtained by the MD Anderson treating physicians from patients with localized ($T_1$–$T_2$, $N_0$, $M_0$) urothelial carcinoma who were candidates for radical cystectomy to receive two doses of anti-CTLA-4 antibody (ipilimumab) before undergoing surgery as per

MD Anderson Cancer Center clinical trial protocol 2006-0080, which was approved by the MD Anderson Cancer Center Institutional Review Board. Six patients completed analyses for safety at the 3 mg kg$^{-1}$ per dose before six additional patients were enroled to receive 10 mg kg$^{-1}$ per dose. Surgery was scheduled on or about week 7 of the protocol. Pre- and post-α-CTLA-4-treated bladder tumour tissues were fixed by immersion in 10% (vol/vol) neutral buffered formalin solution. Fixed tissues were embedded in paraffin and transversely sectioned. Sections of 4 μm were subject to hematoxylin and eosin staining or IHC staining with primary antibodies against CD4 (Novocastra, CD4-368-L-A, 1:80), CD8 (Thermo Scientific, MS-457-S, 1:25), PD-L1 (Cell Signaling, 13684S, 1:100) or PD-1 (Epitomicsm, Abcam, ab137132, 1:250). IHC staining with PD-1 and PD-L1 staining has been described previously[55–57]. Staining from primary antibody was detected with biotinylated secondary antibodies, followed by peroxidase-conjugated avidin/biotin and 3,3'-diaminobenzidine substrate (Leica Microsystem). All IHC slides were scanned and digitalized using a scanscope system (Scanscope XT, Aperio Technologies), and quantitative analyses of IHC staining were conducted using the image analysis software provided (ImageScope-Aperio/Leica). Five random areas (at least 1 mm$^2$ each) were placed within the tumour area using customized algorithm for each specific marker for density (the total number of positive cells per 1 mm$^2$ area), per cent or h-score quantification.

**Statistical analysis.** All experiments were repeated for 2–3 times. Results were expressed as mean ± s.e.m. Data were analyzed using a two-sided Student's t-test or one-way ANOVA after confirming their normal distribution. The log-rank test was used to analyze data from in the survival experiments. All analyses were performed using Prism 5.0 (GraphPad Software, Inc.) and $P < 0.05$ was considered statistically significant.

**Data availability.** All relevant data are available within the manuscript or from the authors upon reasonable request.

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

## Acknowledgements

We thank other members in the Sharma and Allison lab for constructive discussions. We would like to acknowledge the Immunotherapy Platform at MD Anderson Cancer Center for assistance with obtaining appropriate samples from patients on clinical studies. P.S. and J.P.A. are members of the Parker Institute for Cancer Immunotherapy at M. D. Anderson Cancer Center. This study was supported by the SU2C-CRI-AACR Dream Team Grant in Cancer Immunotherapy (P.S. and J.P.A.), NCI/NIH 1-R01 CA1633793-01 (P.S.), and Cancer Prevention Research in Texas Individual Investigator grant, RP120108 (P.S.).

## Author contributions

L.Z.S. designed and did the experiments with cells and mice, analyzed data and wrote the manuscript; F.T. designed and did the experiments with cells and mice, analyzed data and contributed to writing the manuscript; B.G., J.C. and L.X. did the experiments with cells and mice; J.B. did IHC, data analysis of human bladder tumour samples and contributed to writing the manuscript; J.P.A. contributed to overall data analyses and manuscript construction; S.K.S. and J.G. contributed to clinical data collection and analysis; PS was responsible for overall experimental design and supervision for laboratory studies, data analyses and manuscript editing.
