## [Peer Review File · Nature Communications]

Reviewers' comments:

Reviewer #1:Tumor immunology (T-Cells)

(Remarks to the Author):

In this study, Drs. Sharma and Allison and colleagues elucidate mechanisms of tumor rejection after therapy with a combination (combo) of α -CTLA4/ α PD-1 Abs (i.e., the mechanisms underlying "immune checkpoint blockade.") The authors report that concomitant blockade of CTLA-4 and PD-1 upregulated T cell activity and synergistically eliminated MB49 murine bladder cancer by engaging the IL-7/IL-7R and IFN- γ /IFN- γ R signaling pathways.

This is an expertly performed study by a highly experienced team of investigators. The study design is logically developed and well rationalized. The mechanistic underpinnings of antitumor effects of the combo therapy in a murine model of bladder cancer are convincingly demonstrated and clearly presented.

The novel aspect of this work is the demonstration of an intricate interdependent relationship between IFN- γ signaling and IL-7 signaling following the combo therapy that leads to expansion of memory T cells and inhibition of tumor growth.

The authors begin by showing that the PD-1/PD-L1 pathway is a primary suppressive pathway in human as well as murine bladder cancer. This is likely to be true of many but perhaps not all solid tumors. Likewise, the observed "synergy" between α -CTLA-4 and α -PD-1 may hold for many but not all tumors, especially if T cell infiltration in the TME is weak or scarce. It is also questionable whether the combo effects are truly synergistic or only additive?? And intra-tumoral T cells appear to be a critical requirement to success. Hence, the authors are obliged to somewhat temper enthusiasm for recommending the combo therapy for boosting antitumor efficacy in all human tumors, as implied in this manuscript.

The only component of the manuscript directly addressing human cancer is presented in Figure 1 and relates to IHC and flow analyses of TIL or tumor cells in patients with localized bladder carcinoma. The rest of the manuscript deals with mouse data. Thus, there is some concern about extending the mechanistic model presented in SFig 5B to human tumors.

In respect to Fig 1E, G and H, how is the density of positive cells determined? The Methods do not indicate this, and in view of large (not surprising) spread of data points, the analytical method should be described and "density" defined in the legend.

Reviewer #2:Immunotherapy(T-Cells memory)

(Remarks to the Author):

In this manuscript, Shi et al. investigate mice with bladder cancer under treatment with aCTLA4, aPD1 or the combination thereof. The latter was clearly superior in the therapeutic efficacy, and showed synergy, as TILs were enriched and polyfunctional. Enhanced percentages and function of memory T cells were detected. These improved T cell responses depended on IL-7 and IFN γ , leading the authors to conclude that the "Interdependence between IL-7 and IFN γ signaling in T cells controls tumor eradication by combination therapy of α CTLA4 and α PD1".

The paper shows a series of well thought and nicely performed experiments, showing convincing data. The rationale is sound, particularly the point of enhanced memory T cell responses is likely meaningful, as the efficiency of immunotherapy against cancer likely depends on the robust generation of memory T cell responses.

However, the data are to some degree overinterpreted, in part because it remains unclear where and how IL-7 and IFN γ signaling is important.

While the study identifies IL-7ra and IFN γ as key players in responses to dual checkpoint blockade, it does not determine which cells mediate the therapeutic effects depending on IL-7ra and IFN γ , with the exception of the adoptive transfer experiment using knockout T cells shown in Fig. 6C. Unfortunately, the authors show very limited data using this approach. At what time point did the authors assess the shown tumor growth? How was its kinetic? How many mice rejected the tumor? Did the authors characterize the TME and immune cells in this experiment and what did they find?

In view of the scarce information in this regard, most of the data in this paper do not exclude involvement of other cells known to depend on IL-7 and IFN γ , beyond tumor antigen specific CD4 and CD8 effector and memory cells. Yet, the interpretation and discussion is largely focused on the role of the latter. This would require more data from experiments where IL-7 and IFN γ pathways are specifically altered on tumor antigen specific CD4 and CD8 effector and memory T cells specific.

Also, the data does not allow to conclude that IFN γ signaling directly modulates IL-7ra expression in TILs. The literature should be cited with regard to regulators of IL-7ra expression.

For concluding that certain cell populations (such as memory T cells in Figure 5) are truly increased, one must determine and show absolute numbers, not only the percentages. The authors don't indicate "%", but presumably these are percentages (?).

Did the authors observe increased memory T cells in TIL?

How did the authors identify tumor cells and immune cells in the IHC studies shown in Figure 1? What types of immune cells are these? Did they do co-stains or is this assessment solely done based on H&E stains, and thus of limited use?

The experiments with GK1.5 antibody deplete all CD4 T cells, also those of other polarizations than Th1 cells, including Tregs. This point should be discussed.

With the stimulation with antigens and dendritic cells, did the authors also observe Ki-67+ T cells,

and/ or IL-2 or IL-2/IFN γ dual producing CD8 T cells?

It would be interesting to have information on the role of IL-7a in the other therapies, particularly monotherapy with aCTLA-4.

IL-7 has been used in clinical trials, therefore I suggest to revise this statement: "no application of IL-7 in clinical trials has been reported".

Minor points:

Even though the past achievements of this group are outstanding, it might be justified to also cite pioneering work and papers on CTLA4 and PD1 from other researchers, e.g. the first in vivo demonstrations with knockout mice showing the seminal roles of CTLA4.

In the sentence "Currently, immune checkpoint therapy is considered as a standard treatment for patients with cancer", the authors may specify which types of cancer. Alternatively, the sentence may be rephrased by avoiding the word "standard" which currently still only applies to few cancer types.

Some typos, e.g. ...mechanisms remains to...

Reviewer #3: Immune checkpoint therapy
(Remarks to the Author):

This is an interesting manuscript describing the underlying mechanistic effects of combined therapy with anti-PD-1 and anti-CTLA-4 antibodies in a mouse model and in human biopsies of bladder cancer. The mouse model provides phenotypic changes similar to the ones observed in patient-derived biopsies, justifying the use of the model to analyze how this combination exerts its remarkable combinatorial antitumor effects. The manuscript provides compelling evidence and the conclusions are well supported by the data.

Specific comments that could be addressed in the discussion:

Why did the authors choose to study IL-7 signaling in particular? Would it be the same if the testing would be done in hosts that lack signaling through other cytokine gamma-chain receptors?

The studies with IFN γ R1 knock out could be interpreted as having an effect on either the T cells or other hematopoietic lineage cells that traffic to tumors, such as macrophages. The authors performed adoptive cell transfer studies that suggest a major effect of lack of IFN γ R1 in T cells to mediate the antitumor response of the combined immunotherapy. But this experiment does not rule out that lack of IFN γ R1 would result in decreased interferon signaling in macrophages and lack of adaptive PD-L1 expression on these cells.

Reviewers' comments:

Reviewer #1: Tumor immunology (T-Cells)
(Remarks to the Author):

In this study, Drs. Sharma and Allison and colleagues elucidate mechanisms of tumor rejection after therapy with a combination (combo) of α -CTLA4/ α PD-1 Abs (i.e., the mechanisms underlying "immune checkpoint blockade.") The authors report that concomitant blockade of CTLA-4 and PD-1 upregulated T cell activity and synergistically eliminated MB49 murine bladder cancer by engaging the IL-7/IL-7R and IFN- γ /IFN- γ R signaling pathways.

This is an expertly performed study by a highly experienced team of investigators. The study design is logically developed and well rationalized. The mechanistic underpinnings of antitumor effects of the combo therapy in a murine model of bladder cancer are convincingly demonstrated and clearly presented.

The novel aspect of this work is the demonstration of an intricate interdependent relationship between IFN- γ signaling and IL-7 signaling following the combo therapy that leads to expansion of memory T cells and inhibition of tumor growth.

We thank this reviewer for the positive comments.

1) The authors begin by showing that the PD-1/PD-L1 pathway is a primary suppressive pathway in human as well as murine bladder cancer. This is likely to be true of many but perhaps not all solid tumors. Likewise, the observed "synergy" between α -CTLA-4 and α -PD-1 may hold for many but not all tumors, especially if T cell infiltration in the TME is weak or scarce. It is also questionable whether the combo effects are truly synergistic or only additive?? And intra-tumoral T cells appear to be a critical requirement to success. Hence, the authors are obliged to somewhat temper enthusiasm for recommending the combo therapy for boosting antitumor efficacy in all human tumors, as implied in this manuscript.

We agree with this Reviewer's accurate assessment that effects between α -CTLA-4 and α -PD-1 may be "additive" as opposed to "synergistic" in all settings, and this may be true in many but not all solid tumors. Correspondingly, we have rephrased the following statements in the revision.

In the first paragraph of "Introduction", we specify the types of cancers that have been approved by FDA for treatment with immune checkpoint blockade. "Currently, immune checkpoint therapy is considered as a standard treatment for patients with some types of cancer including advanced melanoma, non-small cell lung cancer, and metastatic kidney cancer."

We also revised our manuscript to clearly state that therapeutic effects of combination therapy may be synergistic or additive. For example, in our discussion, we stated

“Mechanistically, we demonstrated that combination therapy synergistically or additively induced greater immune responses ...”.

2) The only component of the manuscript directly addressing human cancer is presented in Figure 1 and relates to IHC and flow analyses of TIL or tumor cells in patients with localized bladder carcinoma. The rest of the manuscript deals with mouse data. Thus, there is some concern about extending the mechanistic model presented in SFig 5B to human tumors.

With the amount of human data that we presented in this manuscript, we understand this Reviewer’s concern about extending our mechanistic model to human tumors. We have revised our discussion to address this concern, as follows,

“It remains to be tested if this interdependent loop between IFN- γ and IL-7 signaling pathways can be extended to human tumors. In our future studies, we propose to analyze IL-7R α expression in human tumors treated with immune checkpoint therapy. Given the considerable similarities between human bladder tumors and murine MB49 bladder tumors, a recent report of IFN- γ upregulation in patients treated with immune checkpoint blockade³⁷, and the essential role of IL-7 and IFN- γ signaling in T cell survival and function, we reason such an interdependent loop likely exists to mediate beneficial effects of combination therapy in human tumors.”

3) In respect to Fig 1E, G and H, how is the density of positive cells determined? The Methods do not indicate this, and in view of large (not surprising) spread of data points, the analytical method should be described and "density" defined in the legend.

We apologize for this oversight and have added details regarding the IHC studies in Materials and Methods. Density is defined as “the total number of positive cells per 1mm² area”.

Reviewer #2: Immunotherapy (T-Cells memory)
(Remarks to the Author):

In this manuscript, Shi et al. investigate mice with bladder cancer under treatment with α CTLA4, α PD1 or the combination thereof. The latter was clearly superior in the therapeutic efficacy, and showed synergy, as TILs were enriched and polyfunctional. Enhanced percentages and function of memory T cells were detected. These improved T cell responses depended on IL-7 and IFN γ , leading the authors to conclude that the "Interdependence between IL-7 and IFN γ signaling in T cells controls tumor eradication by combination therapy of α CTLA4 and α PD1".

The paper shows a series of well thought and nicely performed experiments, showing convincing data. The rationale is sound, particularly the point of enhanced memory T cell responses is likely meaningful, as the efficiency of immunotherapy against cancer likely

depends on the robust generation of memory T cell responses.

We appreciate this Reviewer's positive evaluation of our study.

However, the data are to some degree overinterpreted, in part because it remains unclear where and how IL-7 and IFN γ signaling is important.

*We have taken serious efforts to precisely discuss our findings in this revision. Given our data with in vivo depletion of CD4 and CD8 T cells, the correlation between the abolished therapeutic value of combination therapy and attenuated T cell-mediated immune response in *Ifngr1^{-/-}* and *Il7r^{-/-}* mice, our newly-acquired data of adoptive transfer experiments using WT, *Ifngr1^{-/-}* and *Il7r^{-/-}* total T cells (Fig. 6C-E, Fig. S6A-G), as well as the nature of α -CTLA-4 and α -PD-1 therapy (primarily functioning through T cells), we would like to argue that IL-7 and IFN- γ signaling in T cells is critical for combination therapy (which addresses the reviewer's question of **where**), by controlling T cell survival, infiltration, and effector functions (which addresses the Reviewer's question of **how**).*

While the study identifies IL-7ra and IFN γ as key players in responses to dual checkpoint blockade, it does not determine which cells mediate the therapeutic effects depending on IL-7ra and IFN γ , with the exception of the adoptive transfer experiment using knockout T cells shown in Fig. 6C. Unfortunately, the authors show very limited data using this approach. At what time point did the authors assess the shown tumor growth? How was its kinetic? How many mice rejected the tumor? Did the authors characterize the TME and immune cells in this experiment and what did they find?

*We have included much more new data from the adoptive transfer experiments with KO T cells in this revision. We also revised our manuscript to present the kinetic tumor growth data from the CD45.1 adoptive transfer experiments. Interestingly, while growth of tumors treated with combination therapy appears to be comparable in the early stage of all cohorts, we later observed that tumors regressed in mice that received WT total T cells; however, tumors continued to grow in mice that received either *Il7r^{-/-}* or *Ifngr1^{-/-}* total T cells (Fig. 6C), directly indicating an indispensable role of IL-7 and IFN- γ signaling in T cells in the efficacy of combination therapy. Furthermore, using this adoptive transfer system, we identified combination therapy-improved mouse survival also depends on IL-7 and IFN- γ signaling in T cells (Fig. 6E).*

*To explore how IL-7 and IFN- γ signaling affect T cell proliferation, infiltration into tumors, and function in our adoptive transfer system, we labeled purified WT, *Il7r^{-/-}* or *Ifngr1^{-/-}* total T cells (congenically marked by CD45.2) with CFSE prior to the adoptive transfer. Lack of IFN- γ R1 or IL-7R α in CD4 T cells (Fig. S6A) and CD8 T cells (Fig. S6B) appears not to impact their proliferation, evidenced by comparable dilution of CFSE. However, there is a significant reduction of T cells infiltrated into tumors in the absence of IFN- γ R1 or IL-7R α , with the latter exerting a much stronger effect (Fig. S6C&D). Similar to what we observed in *Il7r^{-/-}* mice, very few *Il7r^{-/-}* T cells infiltrated into the tumors, preventing further accurate analysis on intratumoral T_{reg} abundance and*

effector function. On the other hand, a decent number of $Ifngr1^{-/-}$ $CD4^{+}$ T cells could be detected in the tumor site, which contained significantly more T_{reg} (Fig. S6E) and reduced ratios of effector T cells to regulatory T cells (Fig. S6F), as compared to tumor-infiltrating donor WT counterparts. Moreover, $IFN-\gamma$ -producing $CD4^{+}$ TILs were also significantly reduced in the absence of $IFN-\gamma R1$, although this reduction did not reach the statistical significance in $CD8^{+}$ TILs (Fig. S6G). Taken together, these results indicated $IL-7$ and $IFN-\gamma$ signaling in T cells is important for controlling tumor growth and improving survival likely by regulating T cell effector function and infiltration into tumor.

In view of the scarce information in this regard, most of the data in this paper do not exclude involvement of other cells known to depend on $IL-7$ and $IFN\gamma$, beyond tumor antigen specific $CD4$ and $CD8$ effector and memory cells. Yet, the interpretation and discussion is largely focused on the role of the latter. This would require more data from experiments where $IL-7$ and $IFN\gamma$ pathways are specifically altered on tumor antigen specific $CD4$ and $CD8$ effector and memory T cells specific.

As described above, we provide additional data in this revision to substantiate a key role of $IFN-\gamma$ and $IL-7$ signaling in T cells for mediating antitumor effects of combination therapy. To further address this, we adoptively transferred total WT, $Ifngr1^{-/-}$, or $Il7r^{-/-}$ T cells into $Rag-1$ mice, which lack T and B cells. Therefore, the only T cells present are those transferred and innate immune cells such as macrophages have intact $IFN-\gamma$ and $IL-7$ signaling. This system allows us to unequivocally evaluate if $IL-7$ and $IFN-\gamma$ signaling in T cells are required for tumor suppression by combination therapy. Compellingly, we found that tumor growth was only inhibited by combination therapy when T cells contain intact $IL-7$ and $IFN-\gamma$ signaling (Fig. 6D), pointing to a critical role of $IFN-\gamma$ and $IL-7$ signaling in T cells but not other immune cells in driving antitumor effects of combination therapy. We agree that additional experiments with antigen-specific T cells would strengthen our data but there are no antigen-specific models with MB49 at this point and the defective generation of memory T cells in the $Il7r^{-/-}$ mice would make it extremely difficult to obtain enough KO $CD4$ and $CD8$ memory T cells for further analysis.

Also, the data does not allow to conclude that $IFN\gamma$ signaling directly modulates $IL-7\alpha$ expression in TILs. The literature should be cited with regard to regulators of $IL-7\alpha$ expression.

We agree with this reviewer that $IFN-\gamma$ signaling in stromal cells could directly or indirectly modulate $IL-7\alpha$ expression on TILs upon combination therapy. In our original submission, we discussed the existence of a mutually-regulating loop between $IFN-\gamma$ signaling and $IL-7$ signaling in other different experimental contexts. Per this reviewer's suggestion, we revised our discussion by citing more relevant literatures, shown below.

“In human intestinal epithelial cells, $IFN-\gamma$ can induce $IL-7$ gene expression via binding of its downstream target IFN regulatory factor-1 ($IRF-1$) to an IFN -stimulated response element ($ISRE$) located in the 5' upstream region of the $IL-7$ gene⁴⁹. In addition,

*deficiency of IFN- γ R1 completely diminishes the induction of atypical IL-7R α ⁺ progenitors in bone marrow by acute malaria infection, indicating an essential role of IFN- γ signaling in mediating IL-7R α on bone marrow cells⁵⁰. Further, a recent report demonstrated that IL-12 conditioning, potentially by inducing IFN- γ production, led to increased IL-7R α expression on polarizing Tc1 effector cells, associated with enhanced in vivo persistence and antitumor efficacy of CD8⁺ T cells upon adoptive transfer⁴⁶. Vice versa, an early report suggests that IL-7R signaling contributes to the maintenance of IFN- γ -producing Th1 effector cells during chronic *L. major* infection⁵¹. Thus, our finding corroborates previous reports and reveals for the first time an interactive loop of IFN- γ and IL-7 signaling pathways in TILs that dictates antitumor activity of combination therapy (Fig. S7C)."*

For concluding that certain cell populations (such as memory T cells in Figure 5) are truly increased, one must determine and show absolute numbers, not only the percentages. The authors don't indicate "%", but presumably these are percentages (?).

Thanks for pointing this out. Yes, the numbers in the original figure indicate "% of memory cells". In the revision, we have provided the absolute numbers (Fig. 5A&B) as well, which are also significantly increased upon combination therapy.

Did the authors observe increased memory T cells in TIL?

As known, memory T cells, when formed during the contraction phase, re-express CD62L and CCR7 to migrate to secondary lymphoid organs, which was why we analyzed spleens from mice with retracting tumors. The regressing tumors did not have sufficient cells for accurate assessment of memory T cell subsets.

How did the authors identify tumor cells and immune cells in the IHC studies shown in Figure 1? What types of immune cells are these? Did they do co-stains or is this assessment solely done based on H&E stains, and thus of limited use?

Pathology review of H&E staining is the standard method used by clinical pathologists for distinguishing tumor cells from immune cells. Our pathologists were able to clearly distinguish tumor cells from immune cells. Furthermore, we relied on single-staining IHC studies to determine whether PD-1 and PD-L1 were expressed on tumor cells vs immune cells. IHC studies with antibodies against PD-1 and PD-L1 have been established as previously published and we have added these references to our revised manuscript. We did not perform co-stains for our studies due to the limited amount of tumor tissues.

The experiments with GK1.5 antibody deplete all CD4 T cells, also those of other polarizations than Th1 cells, including Tregs. This point should be discussed.

Thanks for pointing this out. We reason both enhanced T_{eff} function and reduced T_{reg} abundance contribute to the therapeutic effects of combination therapy. Per this Reviewer's suggestion, we have added this sentence to our discussion.

“Since GK1.5 antibody depletes total CD4⁺ T cells, including T_{eff} as well as T_{reg}, we could not distinguish their respective contributions to therapeutic benefits of combination therapy.”

With the stimulation with antigens and dendritic cells, did the authors also observe Ki-67+ T cells, and/ or IL-2 or IL-2/IFN γ dual producing CD8 T cells?

Our new data indicate that intratumoral CD8⁺ TILs also exhibit increased IL-2⁺ IFN- γ ⁺ dual producers (Fig. S3A), when stimulated with irradiated MB49 tumor cells and splenic DC cells.

IL-7 has been used in clinical trials, therefore I suggest to revise this statement: "no application of IL-7 in clinical trials has been reported".

We truly appreciate the reviewer for this point. In this revision, we have taken out the sentence “no application of IL-7 in clinical trials has been reported” and provided a more detailed discussion about the preclinical and clinical applications of IL-7 as an immunotherapeutic, as follows,

“Theoretically, targeting IL-7 signaling should hold greater promises than targeting IL-2 signaling, due to the selective expansion of effector and memory T cell subsets by IL-7 without expanding T_{reg}³⁹, in contrast to undifferentiated expansion of both effector T cells and T_{reg} by IL-2. Recently, it was reported that CAR-engrafted Epstein-Barr-Virus-specific CTLs overexpressing IL-7R α ³⁹ has yielded some successes in preclinical studies. In accordance with this, preclinical application of IL-7 yields substantial antitumor activity using different murine tumor models by expanding effector CD4⁺ and CD8⁺ T cells^{40, 41, 42}. However, two phase I clinical trials of IL-7 in patients with cancer failed to generate overt anti-tumor effects, either as a standalone treatment⁴³ or in combination with vaccination⁴⁴, despite the expected expansion of CD4⁺ and CD8⁺ T cells in IL-7 treated patients⁴³. Given the dual roles of IL-7 in promoting T_{eff} survival and expansion^{40, 43} as well as in suppressing T_{reg} function⁴¹, further explorations of combining immune checkpoint blockade with IL-7 supplementation could potentially lead to improved antitumor therapy, as our data suggested.”

Minor points:

Even though the past achievements of this group are outstanding, it might be justified to also cite pioneering work and papers on CTLA4 and PD1 from other researchers, e.g. the first in vivo demonstrations with knockout mice showing the seminal roles of CTLA4.

We have cited the following two original papers:

- 1. Elizabeth A. Tivol^{*}, Frank Borriello^{*}, A.Nicola Schweitzer^{*}, William P. Lynch^{*}, Jeffrey A. Bluestone[†], Arlene H. Sharpe^{*} Loss of CTLA-4 leads to massive lymphoproliferation and fatal multiorgan tissue destruction, revealing a critical*

- negative regulatory role of CTLA-4. Immunity. 3(5): 541-547.*
2. *Ishida Y, Agata Y, Shibahara K, Honjo T. Induced expression of PD-1, a novel member of the immunoglobulin gene superfamily, upon programmed cell death. EMBO J. 1992 Nov;11(11):3887-95.*

In the sentence "Currently, immune checkpoint therapy is considered as a standard treatment for patients with cancer", the authors may specify which types of cancer. Alternatively, the sentence may be rephrased by avoiding the word "standard" which currently still only applies to few cancer types.

We have changed the sentence to "Currently, immune checkpoint therapy is considered as a standard treatment for patients with some types of cancer including advanced melanoma, non-small cell lung cancer, and metastatic kidney cancer."

Some typos, e.g. ...mechanisms remains to...
Thanks for pointing this out. We have eliminated this typo.

Reviewer #3: Immune checkpoint therapy
(Remarks to the Author):

This is an interesting manuscript describing the underlying mechanistic effects of combined therapy with anti-PD-1 and anti-CTLA-4 antibodies in a mouse model and in human biopsies of bladder cancer. The mouse model provides phenotypic changes similar to the ones observed in patient-derived biopsies, justifying the use of the model to analyze how this combination exerts its remarkable combinatorial antitumor effects. The manuscript provides compelling evidence and the conclusions are well supported by the data.

Specific comments that could be addressed in the discussion:

Why did the authors choose to study IL-7 signaling in particular? Would it be the same if the testing would be done in hosts that lack signaling through other cytokine gamma-chain receptors?

We chose to study IL-7 signaling because of its critical role in maintaining naïve and memory T cells homeostasis, as well as activated effector T cells recruitment into the secondary lymphoid organs. Unlike IL-2, another cytokine signaling through the common γ -chain receptor, IL-7 signaling does not lead to preferential expansion of T_{reg} , an important immunosuppressive factor in TME. As a matter of fact, an early report suggests that IL-7 treatment can suppress T_{reg} function. In addition, in contrast to IL-7, IL-2 has minimal effects in promoting expansion of $CD8^+$ T cells. Considering these functional differences between IL-2 and IL-7 and the distinct phenotypes of $Il7r^{-/-}$ (grossly normal) and $Il2r\alpha^{-/-}$ mice (lymphoproliferative disorder, hemolytic anemia, and an

inflammatory bowel disease, starting from week 9), we would like to argue combination therapy could induce different effects in $Il2ra^{-/-}$ mice. However, the early onset of severe autoimmune defects in these mice prevents a testing with combination therapy. It remains to be tested if other common γ -chain cytokines, such as IL-4, IL-9, IL-15, and IL-21, also play a role in combination therapy-induced anti-tumor effects.

According, we added this sentence to our revision: “Further, in addition to IL-2 and IL-7, other cytokines such as IL-4, IL-9, IL-15 and IL-21 share the common γ -chain receptor to emanate their downstream signals. Thus, it will be interesting to explore if these cytokine signaling pathways also play a role in this process.”

The studies with IFN γ R1 knock out could be interpreted as having an effect on either the T cells or other hematopoietic lineage cells that traffic to tumors, such as macrophages. The authors performed adoptive cell transfer studies that suggest a major effect of lack of IFN γ R1 in T cells to mediate the antitumor response of the combined immunotherapy. But this experiment does not rule out that lack of IFN γ R1 would result in decreased interferon signaling in macrophages and lack of adaptive PD-L1 expression on these cells.

As described above, we conducted new experiments by adoptively transferring total WT, $Ifngr1^{-/-}$, or $Il7r^{-/-}$ T cells into $Rag-1$ mice, which lack mature T and B cells. Therefore, in this system, the only T cells present are those transferred and innate immune cells such as macrophages have intact IFN- γ and IL-7 signaling. This system allows us to unequivocally evaluate if IL-7 and IFN- γ signaling in T cells are required for tumor suppression by combination therapy. We found that tumor growth was only inhibited by combination therapy when T cells have intact IL-7 and IFN- γ signaling (Fig. 6D), clearly pointing to a critical role of IFN- γ and IL-7 signaling in T cells but not other immune cells in driving antitumor effects of combination therapy.

In addition, we have found that:

- 1. Deletion of T cells completely abrogates the therapeutic benefits of combination therapy, in the presence of all the other immune cells.*
- 2. Detailed analysis of $Ifngr1^{-/-}$ and $Il7r^{-/-}$ mice reveals that the diminished therapeutic effects of combination therapy are well correlated with the attenuated T cell-mediated immune responses.*
- 3. We did not observe significant changes of myeloid immune cells such as macrophages (defined as $CD11b^{+}F4/80^{+}$, data not shown), upon combination treatment.*

Therefore, we concluded that IFN- γ signaling in T cells plays a major role in driving anti-tumor effects of combination therapy. Accordingly, we added this sentence to our discussion,

“Based on our findings of abrogation of therapeutic effects of combination therapy in T cell-depleted mice, correlation of diminished therapeutic efficacy of combination therapy to attenuated T cell-mediated immune responses in $Ifngr1^{-/-}$ and $Il7r^{-/-}$ mice, loss of combination therapy-induced IL-7 α upregulation in $Ifngr1^{-/-}$ TILs, and inefficiency of

adoptive transferred Ifngr1^{-/-} or Il7r^{-/-} total T cells to drive combination therapy-mediated suppression of tumor growth and improved mouse survival, we propose that an intricate network consisting of interdependent IFN- γ and IL-7 signaling pathways (not exclusively) in intratumoral T cells controls anti-tumor immunity of combination therapy.”

REVIEWERS' COMMENTS:

Reviewer #1 (Remarks to the Author):

The revised manuscript, including the newly added data, is substantially improved and its overall impact is greater. The authors have done an excellent job in revising the manuscript and responding fully to the reviewers' comments. The manuscript is acceptable for publication.

Reviewer #2 (Remarks to the Author):

The authors have included significantly more data from adoptive transfer experiments, now providing clear evidence that T cells indeed depend significantly on IL-7R and IFN γ R. Also the other points have been adequately addressed.

Reviewer #3 (Remarks to the Author):

The authors have addressed the concerns.